# Ribosomal Protein S4 X-Linked as a Novel Modulator of MDM2 Stability by Suppressing MDM2 Auto-Ubiquitination and SCF Complex-Mediated Ubiquitination

**DOI:** 10.3390/biom14080885

**Published:** 2024-07-23

**Authors:** Satsuki Ryu, Hiroki Nakashima, Yuka Tanaka, Risa Mukai, Yasuhiro Ishihara, Takashi Tominaga, Takayuki Ohshima

**Affiliations:** 1Faculty of Pharmaceutical Science at Kagawa Campus, Tokushima Bunri University, 1314-1 Shido, Sanuki 769-2193, Japan; s198042@stu.bunri-u.ac.jp (S.R.); s158053@stu.bunri-u.ac.jp (H.N.); tominagat@kph.bunri-u.ac.jp (T.T.); 2Faculty of Science and Engineering, Tokushima Bunri University, 1314-1 Shido, Sanuki 769-2193, Japan; s240381@stu.bunri-u.ac.jp; 3Department of Cell Biology and Molecular Medicine, Rutgers New Jersey Medical School, Newark, NJ 07101-1709, USA; 88mukair@gmail.com; 4Program of Biomedical Science, Graduate School of Integrated Sciences for Life, Hiroshima University, 1-7-1 Kagamiyama, Higashi-Hiroshima 739-8521, Japan; ishiyasu@hiroshima-u.ac.jp

**Keywords:** RPS4X, MDM2, Cullin1, SCF complex, ubiquitination

## Abstract

Mouse double minute 2 (MDM2) is an oncoprotein that is frequently overexpressed in tumors and enhances cellular transformation. Owing to the important role of MDM2 in modulating p53 function, it is crucial to understand the mechanism underlying the regulation of MDM2 levels. We identified ribosomal protein S4X-linked (RPS4X) as a novel binding partner of MDM2 and showed that RPS4X promotes MDM2 stability. RPS4X suppressed polyubiquitination of MDM2 by suppressing homodimer formation and preventing auto-ubiquitination. Moreover, RPS4X inhibited the interaction between MDM2 and Cullin1, a scaffold protein of the Skp1-Cullin1-F-box protein (SCF) complex and an E3 ubiquitin ligase for MDM2. RPS4X expression in cells enhanced the steady-state level of MDM2 protein. RPS4X was associated not only with MDM2 but also with Cullin1 and then blocked the MDM2/Cullin1 interaction. This is the first report of an interaction between ribosomal proteins (RPs) and Cullin1. Our results contribute to the elucidation of the MDM2 stabilization mechanism in cancer cells, expanding our understanding of the new functions of RPs.

## 1. Introduction

The protein mouse double minute 2 (MDM2) is a crucial negative regulator of tumor suppressor p53 [1,2,3]. *MDM2* is rarely lost or mutated in human cancer cells [4]; however, *MDM2* overexpression is frequently observed in various human tumors and cancers including ~30% of human sarcomas [5], and MDM2 promotes cell transformation [6].

Although MDM2 amplification has been extensively studied because of its importance in promoting cancer, the molecular mechanisms underlying increased *MDM2* expression in most tumors remain unclear. It has been reported that MDM2 contains a really interesting new gene (RING) finger domain at the C terminus, which functions as an E3 ubiquitin ligase for target proteins such as p53 and itself to promote protein degradation via the ubiquitin–proteasome-mediated pathway [7,8,9,10,11]. Previous studies have reported that the central acidic domain (AD) of MDM2 (residues 220–300) is also critical for the ubiquitination of p53 [12,13]. A small region of the AD of MDM2 (residues 230–260) participates in intramolecular interactions with the RING domain and stimulates E3 ligase activity [14]. AD has a partially unstructured region that interacts with MDM2 regulators, including ribosomal protein (RP) and alternative reading frame product (ARF) [15,16].

RPs are known for their fundamental structural role inherent to ribosomal assembly, maturation, and function. In addition, their ribosome-independent functions related to cell growth, division, and cell death [17] are greatly appreciated. Over the past decade, more than a dozen RPs have been found to activate the tumor suppressor p53 pathway in response to ribosomal stress. In response to nucleolar stress caused by drug-induced ribosomal RNA (rRNA) deficiency, RP abnormality, nutrient starvation, and cell contact suppression, the RPs RPL11, RPL5, RPL23, and RPS7 are released from the nucleolus into the nucleoplasm and associate with MDM2. Subsequently, they inhibit the E3 ubiquitin ligase activity of MDM2 toward p53, thus promoting p53 stabilization and activation [15,18,19,20,21].

Herein, we conducted the yeast two-hybrid screening assay using the central domain of MDM2, including AD, which is important for binding with MDM2 regulators as bait, and identified RPS4X as a novel binding partner, which is an RP encoded on the X chromosome and a component of the 40S subunit. It has been suggested that RPS4X is associated with several diseases such as colorectal cancer and Turner syndrome [22,23]. However, the physiological functions of RPS4X remain unknown. This study aimed to investigate the physiological significance of the interaction between RPS4X and MDM2.

## 2. Materials and Methods

### 2.1. Cell Culture

HEK-293T (CRL-3216) and HeLa (CCL-2) cells were obtained from the American Type Culture Collection (ATCC). HEK-293T and Hela cells were cultured in Dulbecco’s modified Eagle’s medium (FUJIFILM Wako Pure Chemical, Tokyo, Japan), supplemented with 10% fetal bovine serum (Biowest, Nuaillé, France), 100 U/mL penicillin, and 100 µg/mL streptomycin. These cells were maintained at 37 °C under 5% CO_2_ atmosphere.

### 2.2. Antibodies

Mouse anti-hemagglutinin (HA; TANA2; Medical & Biological Laboratories [MBL], Nagoya, Japan), rat anti-HA-HRP (3F10; Roche Diagnostics, Indianapolis, IN, USA), mouse anti-Myc (9E10; Santa Cruz Biotechnology, Santa Cruz, CA, USA), mouse anti-FLAG (M2; Sigma-Aldrich, Inc., St. Louis, MO, USA), and mouse anti-V5 (M215-7; MBL) antibodies were purchased. Anti-DDDK-tag mAb-horseradish peroxidase-linked (HRP)-DirecT, anti-Myc-tag mAb-HRP-DirecT, anti-HA-tag mAb-HRP-DirecT, and anti-V5-tag mAb-HRP-DirecT were purchased from MBL. Mouse anti-β-actin (MBL) antibody, anti-MDM2 antibody (HDM2-323; Santa Cruz Biotechnology), anti-RPS4X antibody (Proteintech, Rosemont, IL, USA), and control rabbit IgG (561; MBL) were also purchased.

### 2.3. Plasmid Construction

The coding regions for human MDM2 and human RPS4X were isolated from the total RNA derived from testicular tissue by reverse transcription-PCR. Full-length MDM2 and RPS4X were subcloned into the *Eco*R1/*Xho*1 sites of pcDNA3 containing 3× FLAG, 4× V5, 6× Myc, and 5× HA epitope tags at the N-terminus, pAcGFP1-C2, or pmCherry. MDM2 and RPS4X deletion mutants were generated by PCR amplification using pcDNA3-3x FLAG-MDM2 or -RPS4X as templates, respectively. Expression vectors for full-length and deletion mutants of Cullin1 and for HA-tagged ubiquitin were previously described [24,25,26]. The primer sequences used were as follows:

MDM2 full-length, 5′-AAAGAATTCATGTGCAATACCAACATGTCTGTA-3′ (forward), 5′-AAACTCGAGCTAGGGGAAATAAGTTAGCAC-3′ (reverse);

MDM2-A, 5′-AAAGAATTCATGTGCAATACCAACATGTCTGTA-3′ (forward), AAACTCGAGTCAAGAAACCAAATGAGAAGATGAA-3′ (reverse);

MDM2-B, 5′-AAAGAATTCAGACCATCTACCTCATCTAGAAGG-3′ (forward), 5′-AAACTCGAGTCATTTATCTTCAGGAAGCCAA-3′ (reverse); MDM2-C, 5′-AAAGAATTCGGGAAAGATAAAGGGGAAATCTC-3′ (forward), 5′-AAACTCGAGCTAGGGGAAATAAGTTAGCAC-3′ (reverse); RPS4X full-length, 5′-AAAGAATTCATGGCTCGTGGTCCCAAGAAGCATCT-3′ (forward), 5′-AAACTCGAGTCACCCACTGCTCTGTTTGGCCGCCA-3′ (reverse);

RPS4X-N, 5′-AAAGAATTCATGGCTCGTGGTCCCAAGAAGCATCT-3′ (forward), 5′-AAACTCGAGCTATCCTTTTGTGCCCACAAAGATC-3′ (reverse); RPS4X-C, 5′-AAAGAATTCATCCCTCATCTGGTGACTCATG-3′ (forward), 5′-AAACTCGAGTCACCCACTGCTCTGTTTGGCCGCCA-3′ (reverse). Underlined letters in primers denote restriction sites.

### 2.4. Yeast Two-Hybrid Screening Assay

The fragment encoding amino acids 161–340 of MDM2 was used as bait to screen a Mate & Plate Library-Universal Human (normalized; Clontech, Kusatsu, Japan), and the assay was performed using the Matchmaker Gold Yeast Two-Hybrid System from Clontech, according to the manufacturer’s instructions.

### 2.5. Immunoprecipitation and Immunoblotting

HEK-293T cells (1 × 10^6^ cells per 6-cm diameter dish) were transfected with expression plasmids using polyethyleneimine (PEI MAX, linear, molecular weight 40 kDa; Polysciences, Inc., Warrington, PA, USA). After 36 h, the cells were lysed in 1 mL of lysis buffer (25 mM Tris-HCl [pH 8.0], 150 mM NaCl, 1% Nonidet P-40, 1 mM ethylene diamine tetra acetic acid [EDTA], Pefabloc SC Plus inhibitor [Roche Diagnostics], and cOmplete ULTRA tablets [Roche Diagnostics]), and cell debris was removed by centrifugation for 15 min. The cell lysates were incubated with appropriate antibodies for 1 h at 4 °C, and antibody complexes were captured using Protein G-SepharoseTM 4 Fast Flow (GE Healthcare, Buckinghamshire, UK) for 1 h at 4 °C. Protein-bound beads were washed three times with the same buffer, and immunoprecipitants were eluted in sodium dodecyl sulfate (SDS) sample buffer. The samples were separated using SDS–polyacrylamide gel electrophoresis (SDS-PAGE), transferred to a polyvinylidene difluoride membrane (Merck Millipore, Bedford, MA, USA), blocked with 5% skim milk-phosphate-buffered saline containing 0.05% Tween, probed with the indicated antibodies, and developed using chemiluminescence.

### 2.6. In Vivo Ubiquitination Assay

HEK-293T cells were transfected with the indicated plasmids using polyethylenimine “Max” (PEI MAX). After treatment with 40 µM of the proteasome inhibitor MG132 (Peptide Institute Inc., Osaka, Japan) for 4 h, the cells were lysed in 1 mL of lysis buffer (25 mM Tris-HCl [pH 8.0], 150 mM NaCl, 1% Nonidet P-40, 1 mM EDTA, 0.1% SDS, deoxycholic acid sodium salt monohydrate [Nacalaitesque, Kyoto, Japan], Pefabloc SC Plus inhibitor [Roche Diagnostics], and cOmplete ULTRA tablets [Roche Diagnostics]), and cell debris was removed by centrifugation for 15 min. The cell lysates were incubated with antibodies for 1 h at 4 °C, and antibody complexes were captured using Protein G-SepharoseTM 4 Fast Flow beads for 1 h at 4 °C. The beads were washed three times with the same buffer, and immunoprecipitants were eluted. The samples were subjected to SDS-PAGE, followed by immunoblot analysis.

### 2.7. Fluorescence Microscopy and Imaging

HeLa cells were seeded at a density of 1 × 10^6^ cells per 6 cm diameter dish and transfected with expression plasmids using PEI MAX. At 20 h post-transfection, the cells were stained with Hoechst 33342 (FUJIFILM Wako Pure Chemical) for 30 min at 37 °C to label the nuclei before undergoing fluorescence microscopy (IX71; Olympus Life Science, Tokyo, Japan).

### 2.8. Cycloheximide Chase Assay

HEK-293T cells were seeded in 12-well plates (1 × 10^5^ cells/well) and transfected with the appropriate expression plasmids. After 24 h, the cells were treated with 100 µg/mL cycloheximide (FUJIFILM Wako Pure Chemical) to inhibit protein synthesis and chased for the indicated intervals. The harvested cells were analyzed by immunoblotting. The band intensities were measured using a LAS 3000 image analyzer (Fujifilm, Tokyo, Japan).

## 3. Results

### 3.1. RPS4X Interacts with MDM2 in Nucleoplasm

To screen for proteins that interact with the central region of MDM2, we used the yeast two-hybrid system. A screen of 4 × 10^5^ primary transformants of the universal human cDNA library yielded 165 potential positive clones. Of these candidates, sequence analysis revealed several positive clones corresponding to RPS4X. At first, to determine whether RPS4X interacted with MDM2 in human cells, co-immunoprecipitation was performed. The results showed that RPS4X interacted with MDM2 (Figure 1A). Subsequently, to define whether endogenous RPS4X interacted with MDM2 in human cancer cells, we conducted a co-immunoprecipitation assay using HeLa cells. Cell extracts were immunoprecipitated with control IgG or anti-RPS4X antibodies, respectively. As shown in Figure 1B, endogenous MDM2 was detected in the co-immunocomplexes of RPS4X in HeLa cells.

We hypothesized that RPS4X interacts with MDM2 in the nucleoplasm. To test this hypothesis, we analyzed the subcellular localizations of RPS4X and MDM2 in transiently transfected HeLa cells. It was previously reported that MDM2 is expressed primarily in the nucleoplasm, except for nucleolus [27,28]. Similar to these reports, our results showed that GFP-MDM2 alone was distributed throughout the nucleoplasm, except for the nucleolus (Figure 1C, upper-left panels). Furthermore, mCherry-RPS4X alone was localized in the nucleolus (Figure 1C, upper-right panels), which is consistent with previously reported findings in RPL11, RPL5, RPL23, and RPS7 [15,18,19,20,21]. When RPS4X and MDM2 were co-expressed in the cells, RPS4X was distributed outside the nucleolus and matched with MDM2 in the nucleoplasm (Figure 1C, lower panels). Our findings suggested that in human cells, RPS4X is released outside the nucleolus and interacts with MDM2 in the nucleoplasm.

### 3.2. N-Terminus of RPS4X Binds to the Central Domain of MDM2

To identify the RPS4X and MDM2 domains required for the interaction between both proteins, we constructed deletion mutants (Figure 2A,C) and performed immunoprecipitation assays (Figure 2B,D). First, we determined the binding domain of MDM2 to RPS4X. HEK-293T cells were transfected with plasmids expressing FLAG-MDM2-full-length or a series of FLAG-tagged MDM2 deletion mutants (-A, -B, and -C) together with the Myc-tagged form of RPS4X, and the cell extracts were immunoprecipitated with anti-FLAG antibody, followed by immunoblotting with anti-Myc antibody. As shown in Figure 2B, RPS4X was detected in immunocomplexes isolated from the cells expressing full-length MDM2 (MDM2-full) and MDM2-B.

Subsequently, to determine the binding domain of RPS4X with MDM2, HEK-293T cells were co-expressed with V5-MDM2 together with Myc-RPS4X-full-length or a series of Myc-tagged RPS4X deletion mutants (-N, -C; Figure 2C, top panel), and the cell extracts were immunoprecipitated with anti-Myc antibody, followed by immunoblotting with anti-V5 antibody. As shown in Figure 2D, MDM2 was detected in immunocomplexes isolated from the cells expressing RPS4X-full and RPS4X-N, but not in those expressing RPS4X-C; this suggests that MDM2 interacts with the N-terminus of RPS4X. Thus, RPS4X interacts with the central domain of MDM2 via its N-terminus.

### 3.3. RPS4X Prevents the Formation of the MDM2-MDM2 Homodimer and Suppresses MDM2 Ubiquitination

Generally, many E3 ubiquitin ligases, including MDM2, undergo auto-ubiquitination [29,30]. MDM2 homodimerization requires the extreme C-terminus and central AD of MDM2, suggesting that MDM2 homodimers utilize distinct MDM2 domains [31]. As RPS4X interacts with the central domain of MDM2, we postulated that RPS4X might suppress the formation of MDM2 homodimers. To test this hypothesis, HEK-293T cells were co-expressed with Myc-MDM2 and FLAG-MDM2, with or without HA-RPS4X, and the cell extracts were then immunoprecipitated with anti-FLAG antibody and immunoblotted with anti-Myc antibody. As expected, when RPS4X was co-expressed in the cells, the formation of MDM2 homodimers was markedly suppressed (Figure 3A, lane 2 versus lane 3, top panel).

Most intercellular proteins are degraded by the ubiquitin-proteasome system, and the majority of proteasomal substrates are targeted for degradation by conjugation to polyubiquitin chains. We postulated that RPS4X inhibition of MDM2 homodimer formation would affect the ubiquitination level of MDM2. To test this hypothesis, we investigated whether RPS4X regulates the ubiquitination of MDM2. HEK-293T cells were co-transfected with plasmids expressing FLAG-MDM2, HA-ubiquitin, and Myc-RPS4X (Figure 3B). After 24 h, the cells were treated with the proteasome inhibitor MG132 for 16 h. The cell extracts were immunoprecipitated with anti-FLAG antibody, followed by immunoblotting with anti-HA antibody. As shown in Figure 3B (top panel), the ubiquitination level of MDM2 was reduced by RPS4X expression in the cells.

### 3.4. RPS4X Associates with the N-Terminus of Cullin1 through Its C-Terminus and Potently Inhibits MDM2/Cullin1 Complex Formation

A previous study showed that even in mice with an MDM2 mutation that prevents auto-ubiquitination, MDM2 is still degraded by the ubiquitin–proteasome system [32]. In response to this report, the SCF complex was identified as an E3 ubiquitin ligase responsible for the destruction of MDM2 protein [33]. Moreover, MDM2 specifically interacts with Cullin1, which is a key component of the SCF complex [33,34]. Thus, to elucidate the detailed mechanism underlying the suppression of MDM2 ubiquitination by RPS4X, we hypothesized that RPS4X affects the interaction between MDM2 and Cullin1. To examine this hypothesis, HEK-293T cells were co-expressed with V5-MDM2 with (+) or without (−) Myc-Cullin1 and HA-RPS4X. The cell extracts were immunoprecipitated with anti-Myc antibody and immunoblotted with anti-V5 antibody. As shown in Figure 4A, when MDM2 and Cullin1/RPS4X were co-expressed in the cells, the interaction between MDM2 and Cullin1 was significantly suppressed compared with that in cells without RPS4X (lane 2 versus lane 3). Since the interaction between MDM2 and Cullin1 was substantially attenuated by RPS4X, we expected that RPS4X would interact with both MDM2 and Cullin1. To analyze the interaction between RPS4X and Cullin1, co-immunoprecipitation was performed. HEK-293T cells were transfected with plasmids expressing Myc-Cullin1 and FLAG-RPS4X, and the cell extracts were immunoprecipitated with anti-FLAG antibody, followed by immunoblotting with anti-Myc antibody. As shown in Figure 4B, RPS4X interacts with Cullin1.

To determine the interaction domain between RPS4X and Cullin1, deletion mutants of RPS4X were generated (Figure 2C) and the co-immunoprecipitation assay was performed. As shown in Figure 4C, RPS4X was detected in the immunocomplexes isolated from the cells expressing full-length RPS4X (RPS4X-full) and RPS4X-C (Figure 4C, lane 2 and lane 4, respectively), but not RPS4X-N (Figure 4C, lane 3). Therefore, the C-terminus of RPS4X is required for its interaction with Cullin1. To define the Cullin1 binding region of RPS4X in more detail, we constructed the deletion mutants shown in Figure 4D and conducted immunoprecipitation assays (Figure 4E). HEK-293T cells were co-expressed with FLAG-RPS4X and Myc-Cullin1-full-length or a series of Myc-tagged Cullin1 deletion mutants (-A, -B, and -C), and the cell extracts were immunoprecipitated with anti-Myc antibody, followed by immunoblotting with anti-V5 antibody. As shown in Figure 4E, RPS4X was detected in the immune complexes isolated from the cells expressing Cullin1-full and Cullin1-A (Figure 4E, lane 2 and lane 3, respectively). Thus, RPS4X significantly inhibits MDM2/SCF complex formation by binding to the N-terminus of Cullin1 via its C-terminus.

### 3.5. MDM2 Stability Is Enhanced by RPS4X

To confirm whether RPS4X extends the half-life of MDM2 by inhibiting the association between MDM2 and Cullin1, we performed a cycloheximide chase assay. HEK-293T cells were co-expressed with Myc-MDM2 and FLAG-Cullin1 with or without FLAG-RPS4X. Cycloheximide is a translational inhibitor of eukaryotic protein synthesis. As shown in Figure 5A, MDM2 protein levels declined at a slower rate in the cells co-expressing RPS4X. In this result, the half-life of MDM2 seems to differ significantly from previous studies for MDM2 [29,35]. It is well known that the half-life of MDM2 varies widely among cell lines and is also tightly regulated by various post-translational modifications [36]. In view of these reports, the half-life of MDM2 seems to vary greatly depending on the individual experimental conditions. Since the half-life of MDM2 was extended by RPS4X, we expected that the steady-state level of the MDM2 protein would be enhanced by RPS4X. To test this hypothesis, HEK-293T cells were transfected with a plasmid expressing V5-MDM2 and Myc-RPS4X. As shown in Figure 5B, RPS4X overexpression resulted in increased MDM2 protein levels. These results demonstrate that RPS4X enhances the stability of MDM2 by inhibiting the association between MDM2 and Cullin1.

## 4. Discussion

Much evidence suggests that MDM2 is overexpressed in many human tumors and promotes cancer progression and drug resistance [37,38,39,40]. RPs are known to exert tumor-suppressive effects via the RP-MDM2-p53 pathway. However, some RPs with cancer-promoting functions were reported. For instance, RPL34 promotes the proliferation, migration, and invasion of pancreatic and glioma cancer cells, both in vitro and in vivo [41,42]. RPL32 overexpression is associated with poor prognosis in patients with lung cancer [43]. Recently, it has been reported that most rectal colon cancer cell lines express high levels of RPS4X and that the expression levels are negatively correlated with tumor prognosis [22]. A previous study suggested that overexpression of RPS4X is correlated with poor prognosis in not only colon rectal cancer but also intrahepatic cholangiocarcinoma [44]. Additionally, overexpression of MDM2 protein was observed in 38% of intrahepatic cholangiocarcinoma tumors and correlated with the presence of metastases (*p* < 0.01) and advanced tumor stage (*p* < 0.05) [45]. In view of these reports and our findings in the present study, an enhancement in MDM2 protein stability by RPS4X could potentially lead to tumor progression and poor prognosis. Further studies are required to elucidate the functional consequences of the interaction between RPS4X and MDM2 in cancer cells and may contribute to reinforcing the evidence that it could be a biomarker for these cancers, which may help in more effective treatment for patients.

MDM2 is a substrate of the SCF complex, a well-known ubiquitin ligase. The SCF complex is composed of Skp1, Cullin1, F-BOX protein, and Rbx1 and catalyzes the ubiquitination of molecules in the G1/S/G2 phase, such as G1 cyclin and cyclin-dependent kinase (CDK) inhibitor. We found that RPS4X is bound to the N-terminus of Cullin1 (Figure 4E). The N-terminus of Cullin1 binds to Skp1, and Skp1 binds to F-box protein, a substrate recognition subunit. Therefore, RPS4X may affect the binding of Cullin1 and Skp1 and may be widely involved in the degradation of the SCF complex substrate. Recent studies have established a link between RPs and cell cycle progression. RPL3 induces G1 cell cycle arrest because it is involved in regulating the expression and stabilization of p21, a CDK inhibitor [46,47]. RPS13 also inhibits the mRNA expression of the CKD inhibitor p27 and accelerates the G1/S transition [48]. Our results suggest that RPS4X is an RP with the potential to be involved in the cell cycle via a novel mechanism mediated by the SCF complex. Further research is required to elucidate the effect of RPS4X on the regulation of target proteins in the SCF complex, which will contribute to expanding our understanding of the ribosomal-independent functions of RPs.

## Figures and Tables

**Figure 1 biomolecules-14-00885-f001:**
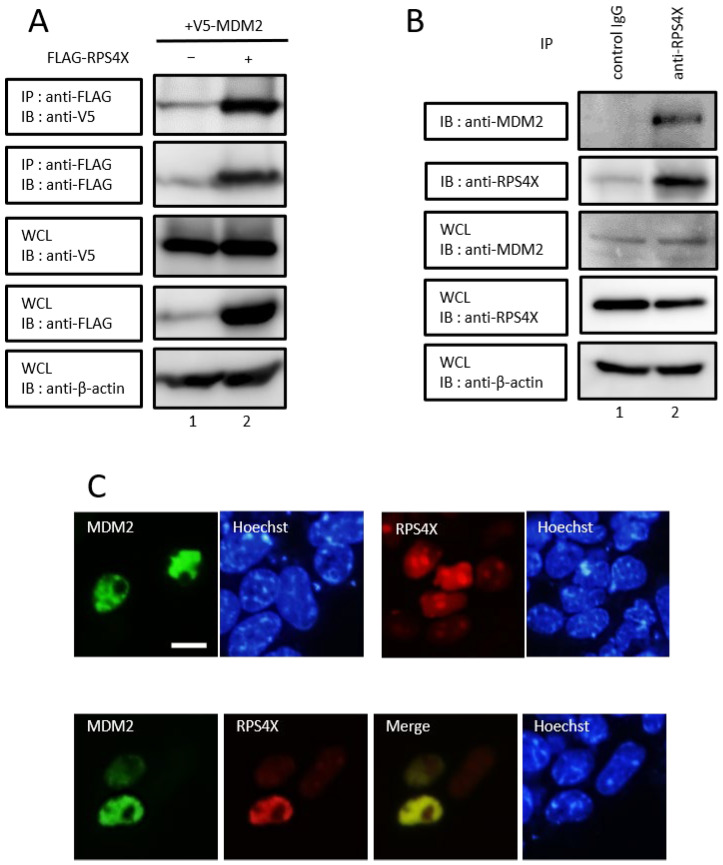
RPS4X interacts with MDM2 in the nucleoplasm. (**A**) HEK-293T cells were co-transfected with 3 µg of plasmid expressing V5-MDM2 with (+) or without (−) 3 µg of plasmid expressing FLAG-RPS4X. After 36 h, the cell extracts were prepared and subjected to immunoprecipitation (IP) using anti-FLAG antibody, followed by immunoblotting (IB) with anti-V5 and anti-FLAG antibodies. Total protein levels in whole-cell lysates (WCLs) were analyzed by IB using anti-FLAG, anti-V5, and anti-β-actin antibodies. (**B**) HeLa cell extracts were prepared and subjected to IP using control rabbit IgG or anti-RPS4X antibodies, followed by IB with anti-MDM2 and anti-RPS4X antibodies. Total protein levels in WCLs were analyzed by IB using anti-MDM2, anti-RPS4X, and anti-β-actin antibodies. (**C**) HeLa cells were transfected with 1 µg of plasmid expressing AcGFP-MDM2 or mCherry-RPS4X, respectively (**top panels**), and were co-transfected with 1 µg of plasmid expressing AcGFP-MDM2 and mCherry-RPS4X (**bottom panels**). After 24 h, the cells were counterstained with Hoechst 33342 (Hoechst) to detect the nuclei and visualized using a fluorescence microscope. Scale bars represent 20 µm. RPS4X, ribosomal protein S4X-linked; MDM2, mouse double minute 2. Please see the original images of Figure 1 in Appendix A.

**Figure 2 biomolecules-14-00885-f002:**
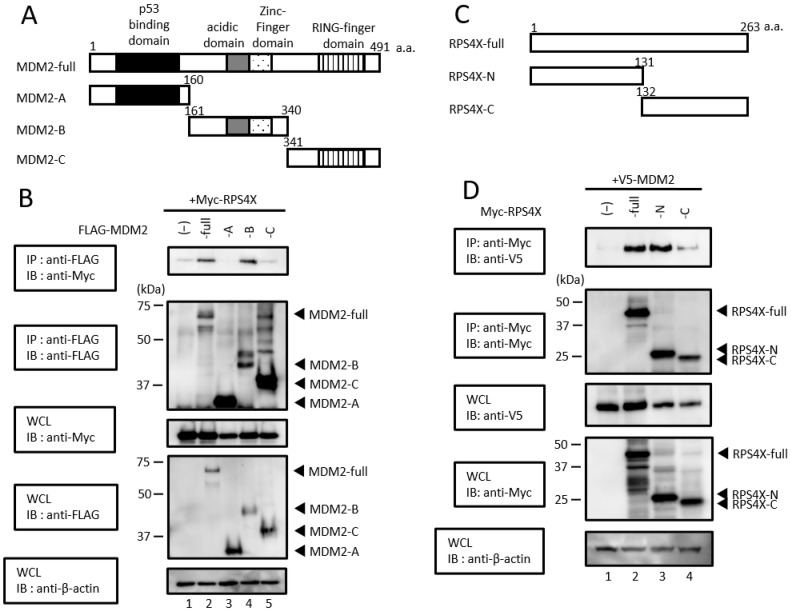
Determination of binding domains between RPS4X and MDM2. (**A**) Schematic diagram of full-length MDM2 (MDM2-full) and deletion mutants (MDM2-A, -B, and -C) used in this study. Characteristic domains of MDM2 are indicated as follows: black box, p53 binding domain; gray box, acidic domain; dot box, Zinc-finger domain; stripe box, RING-finger domain; a.a., amino acid. (**B**) HEK-293T cells were co-transfected with 3 µg of plasmid expressing Myc-RPS4X with (+) or without (−) 3 µg of plasmid expressing FLAG-MDM2-full, -MDM2-A, -MDM2-B, or -MDM2-C. After 36 h, the cell extracts were prepared and subjected to immunoprecipitation (IP) using anti-FLAG antibody, followed by immunoblotting (IB) with anti-Myc and anti-FLAG antibodies. Total protein levels in whole-cell lysates (WCLs) were analyzed by IB using anti-Myc, anti-FLAG, and anti-β-actin antibodies. (**C**) Schematic diagram of full-length RPS4X (RPS4X-full) and deletion mutants (RPS4X-N and -C) used in this study. a.a., amino acid. (**D**) HEK-293T cells were co-transfected with 2 µg of plasmid expressing V5-MDM2 with (+) or without (−) 2 µg of plasmid expressing Myc-RPS4X-full, -RPS4X-N, or -RPS4X-C. After 36 h, the cell extracts were prepared and subjected to IP using anti-Myc antibody, followed by IB with anti-V5 and anti-Myc antibodies. Total protein levels in WCLs were analyzed by IB using anti-V5, anti-Myc, and anti-β-actin antibodies. RPS4X, ribosomal protein S4X-linked; MDM2, mouse double minute 2. Please see the original images of Figure 2 in Appendix A.

**Figure 3 biomolecules-14-00885-f003:**
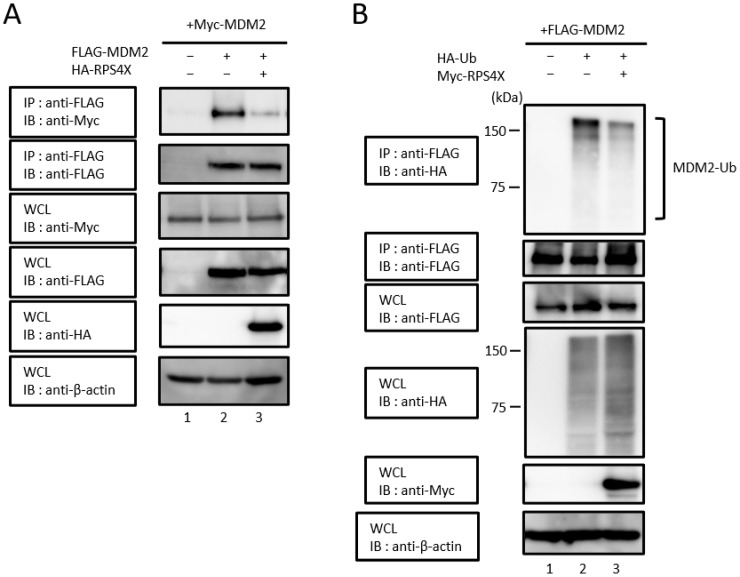
MDM2 homodimer formation and MDM2 ubiquitination are suppressed by RPS4X. (**A**) HEK-293T cells were co-transfected with 1 µg of plasmid expressing Myc-MDM2 with (+) or without (−) 1 µg of plasmid expressing FLAG-MDM2 and 3 µg of plasmid expressing HA-RPS4X. After 24 h, the cells were treated with 20 µM MG132 (a proteasome inhibitor) for 16 h. The cell extracts were prepared and subjected to immunoprecipitation (IP) using anti-FLAG antibody, followed by immunoblotting (IB) with anti-Myc and anti-FLAG antibodies. Total protein levels in whole-cell lysates (WCLs) were analyzed by IB using anti-Myc, anti-FLAG, anti-HA, and anti-β-actin antibodies. (**B**) HEK-293T cells were co-transfected with 2 µg of plasmid expressing FLAG-MDM2 with (+) or without (−) 1 µg of plasmid expressing HA-ubiquitin (Ub) or Myc-RPS4X. After 24 h, the cells were treated with 40 µM MG132 (a proteasome inhibitor) for 16 h. The cell extracts were prepared and subjected to IP using anti-FLAG antibody, followed by IB with anti-HA and anti-FLAG antibodies. Total protein levels in WCLs were analyzed by IB using anti-FLAG, anti-HA, anti-Myc, and anti-β-actin antibodies. RPS4X, ribosomal protein S4X-linked; MDM2, mouse double minute 2. Please see the original images of Figure 3 in Appendix A.

**Figure 4 biomolecules-14-00885-f004:**
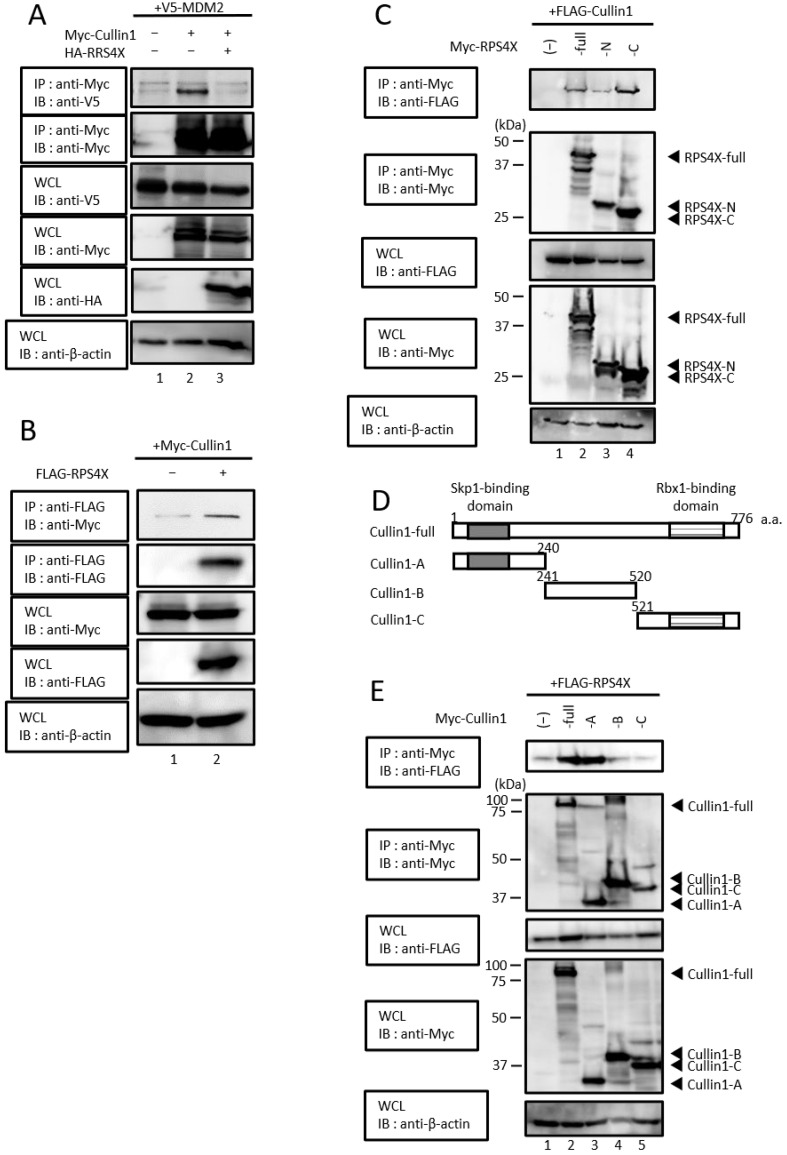
RPS4X binds to Cullin1 and inhibits the interaction between MDM2 and Cullin1. (**A**) HEK-293T cells were co-transfected with 2 µg of plasmid expressing V5-MDM2 with (+) or without (−) 2 µg of plasmid expressing Myc-Cullin1 or HA-RPS4X. After 36 h, the cell extracts were prepared and subjected to immunoprecipitation (IP) using anti-Myc antibody, followed by immunoblotting (IB) with anti-V5 and anti-Myc antibodies. Total protein levels in whole-cell lysates (WCLs) were analyzed by IB using anti-V5, anti-Myc, anti-HA, and anti-β-actin antibodies. (**B**) HEK-293T cells were co-transfected with 2 µg of plasmid expressing Myc-Cullin1 with (+) or without (−) 2 µg of plasmid expressing FLAG-RPS4X. After 36 h, the cell extracts were prepared and subjected to IP using anti-FLAG antibody, followed by IB with anti-Myc and anti-FLAG antibodies. Total protein levels in WCLs were analyzed by IB using anti-Myc, anti-FLAG, and anti-β-actin antibodies. (**C**) HEK-293T cells were co-transfected with 2 µg of plasmid expressing FLAG-Cullin1 with (+) or without (−) 2 µg of plasmid expressing Myc-RPS4X-full, -RPS4X-N, or -RPS4X-C. After 36 h, the cell extracts were prepared and subjected to IP using the anti-Myc antibody, followed by IB with anti-FLAG and anti-Myc antibodies. Total protein levels in WCLs were analyzed by IB using the anti-FLAG, anti-Myc, and anti-β-actin antibodies. (**D**) Schematic diagram of the full-length Cullin1 (Cullin1-full) and deletion mutants (Cullin1-A, -B, and -C) used in this study. Characteristic domains of Cullin1 are indicated as follows: gray box, Skp1 binding domain; horizontal stripes box, Rbx1 binding domain; a.a., amino acid. (**E**) HEK-293T cells were co-transfected with 2 µg of plasmid expressing FLAG-RPS4X with (+) or without (−) 2 µg of plasmid expressing Myc-Cullin1-full, -Cullin1-A, -Cullin1-B, or -Cullin1-C. After 36 h, the cell extracts were prepared and subjected to IP using anti-Myc antibody, followed by IB with anti-FLAG and anti-Myc antibodies. Total protein levels in WCL were analyzed by IB using anti-FLAG, anti-Myc, and anti-β-actin antibodies. RPS4X, ribosomal protein S4X-linked; MDM2, mouse double minute 2. Please see the original images of Figure 4 in Appendix A.

**Figure 5 biomolecules-14-00885-f005:**
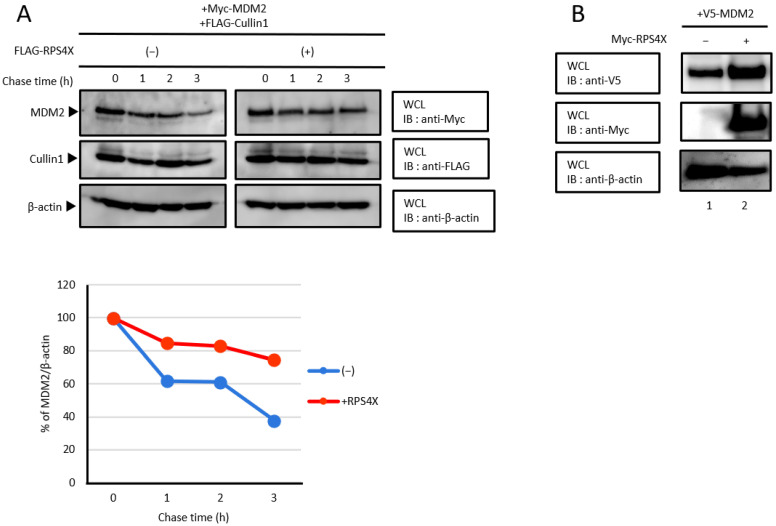
MDM2 stability is enhanced by RPS4X. (**A**) HEK-293T cells were co-transfected with 0.5 µg of plasmid expressing Myc-MDM2 and FLAG-Cullin1 with (+) or without (−) 3 µg of plasmid expressing FLAG-RPS4X. After 24 h, the cells were treated with 100 µg/mL cycloheximide (protein synthesis inhibitor) and chased for the indicated time intervals. The cell lysates were subjected to SDS-PAGE followed by immunoblotting (IB) with anti-Myc antibody, and the MDM2 protein levels are shown. The intensity of each band was quantified by densitometry using ImageJ v1.43 software and graphed. (**B**) HEK-293T cells were co-transfected with 3 μg of plasmid expressing V5-MDM2 with (+) or without (−) 3 μg of plasmid expressing Myc-RPS4X. After 36 h, the cell extracts were subjected to SDS-PAGE, followed by IB with anti-V5, anti-Myc, and anti-β-actin antibodies. RPS4X, ribosomal protein S4X-linked; MDM2, mouse double minute 2. Please see the original images of Figure 5 in Appendix A.

## Data Availability

The data are available on reasonable request from the corresponding author.

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
