# Peer review of "Ribosomal Protein S4 X-Linked as a Novel Modulator of MDM2 Stability by Suppressing MDM2 Auto-Ubiquitination and SCF Complex-Mediated Ubiquitination"

_biomolecules, 2024, doi:10.3390/biom14080885_

Round 1
Reviewer 1 Report
Comments and Suggestions for Authors
The Paper with the title:
Ribosomal protein S4 X-linked as a novel modulator of MDM2 2 stability by suppressing MDM2 auto-ubiquitination and SCF 3 complex-mediated ubiquitination
By :
Satsuki Ryu 1 , Hiroki Nakashima 1 , Yuka Tanaka 2 , Risa Mukai 3 , Yasuhiro Ishihara 4 , Takashi Tominaga 1 and Taka- 5 yuki Ohshima 1,2*
identified the ribosomal protein S4 X linked as a new binding partner of MdM2 by yeast two hybrid assay. The interaction was confirmed by co immunoprecipitation. Immunofluorescence pictures showed interaction in the nucleoplasm. Mutational studies revealed binding of the central part of MDM2 with the N-terminus of RPS4X. Co expression of MDM2 and RPS4X prevents homodimer formation of MDM2 and subsequently reduces the formation of poly-ubiquitinylated MDM2. RPS4X was found to associate with the N-terminus of Cullin1 through its C-terminus and potently inhibits MDM2/Cullin1 formation, by transfection and immune precipitation of the respective proteins. The authors performed an cycloheximide chase to determine the half live of MDM2 with and without RPS4X, and found a longer half life of MDM2 in the presence of RPS4X.
The introduction is well written and gives enough information that also non experts understand the topic. The methods and material section contains all required information to reproduce the experiments. The figures and legends provide all informations to understand them, without reading the whole text. The discussion is concise, but cites all relevant literature. Unfortunately, not all conclusions are justified by the presented results.
Major points of criticism:
1. The authors did not present any evidence for the interaction of MDM2 and RPS4X under endogenous expression. The proteins in the publication are always overexpressed. As the concentration is of importance for such interactions, the endogenous expression level is the final proof whether or not these interaction can occur at all in vivo.
2. The authors used HA-tagged Ubiquitin for the experiment in fig. 3B, to show less ubiquitinylation of MDM2. This is not conclusive, as the is no control showing equal expression or transfection of HA-Ubiquitin. The cell is full of Ubiquitin, so there is no need to transfect it. Anti-Ubiquitin antibodies are commercially available. The original figure 3B panel 2 should also show polyubiquitinylated MDM2 in the upper part, as IP flag followed by WB anti flag must stain all present MDM2, also the polyubiquitinylated. In this figure the highest signal in the upper part (shmear) is in lane 1, and lane 3 shows more signal than lane 2, so potentially a different result as with the anti HA antibody. The same impression one might get by looking at 3B panel 3, WB of MDM2 in the lysate. The shmear is darkest in lane 1, lane 2 and 3 are similar.
3. The results presented in section 3.4. are not conclusive. There is a lot of cullin in the cell, there is no need to transfect it. The authors did not address if the cullin, they transfected, is incorporated in to the SCF complex. It is possible that the interaction, ( competition ) occurs only with free cullin and is not relevant for the SCF complex. One could immunoprecipitate a different subunit of the SCF complex, and detect co immunoprecipitated proteins like, cullin and MDM2, as monomeric cullin will not support any ubiquitin chain formation.
4. The sentence staring in lane 24: “ RPS4X expression in cells enhanced the steady-state level of 24 MDM2 protein. “ is not justified by the results presented. The experiment presented in section 3.5 is very convincing, but it shows a difference in half life, rather than in steady-state level. The values for MDM2 at time 0 with and without RPS4X look very similar. Also the amount of MDM2 in fig 3 B panel 2 and 3 does not show a serious difference with and without RPS4X, the WB on the lysate appears to have even less MDM2 with RPS4X. It seams that the difference in degradation is somehow compensated by higher expression. This may be due to the over expression of the proteins. The half life reported also differs significantly from other reported half life’s of MDM2. Please see: https://www.jbc.org/article/S0021-9258(19)34705-2/pdf; and https://www.ncbi.nlm.nih.gov/pmc/articles/PMC391059/;
These differences may be due to different cell lines and or expression levels. The authors should at least discuss the differences.
Minor points:
There are no molecular weight markers on any of the WB presented. This makes it difficult to judge the quality.
Author Response
We are grateful to reviewer #1 for the critical comments and useful suggestions that have helped us to improve our paper considerably. As indicated in the responses that follow, we have taken all these comments and suggestion into account in the revised version of our paper.
Major points:
- The authors did not present any evidence for the interaction of MDM2 and RPS4X under endogenous expression. The proteins in the publication are always overexpressed. As the concentration is of importance for such interactions, the endogenous expression level is the final proof whether or not these interaction can occur at all in vivo.
Response: We thank the reviewer for this insightful comment. This point was also mentioned by reviewer 2. We investigated the interaction between endogenous MDM2 and RPS4X in HeLa cells by co-immunoprecipitation assay. As a result, endogenous MDM2/RPS4X interaction was also observed in cells. This result was added to Figure 1B as a new data and updated the manuscript.
- The authors used HA-tagged Ubiquitin for the experiment in fig. 3B, to show less ubiquitinylation of MDM2. This is not conclusive, as the is no control showing equal expression or transfection of HA-Ubiquitin. The cell is full of Ubiquitin, so there is no need to transfect it. Anti-Ubiquitin antibodies are commercially available. The original figure 3B panel 2 should also show polyubiquitinylated MDM2 in the upper part, as IP flag followed by WB anti flag must stain all present MDM2, also the polyubiquitinylated. In this figure the highest signal in the upper part (shmear) is in lane 1, and lane 3 shows more signal than lane 2, so potentially a different result as with the anti HA antibody. The same impression one might get by looking at 3B panel 3, WB of MDM2 in the lysate. The shmear is darkest in lane 1, lane 2 and 3 are similar.
Response: We thank the reviewer for these comments. The first reason, we utilized an anti-HA antibody is that this rat monoclonal HA antibody (3F10) is well-known for its very high affinity and extremely high specificity. As the reviewer pointed out, the amount of endogenous ubiquitin is very abundant, and we think WB with anti-Ub antibodies are useful. However, in this study, we used rat monoclonal HA antibody (3F10), one of the most specific antibodies widely used in the research world, because we wanted to analyze changes in polyubiquitination of MDM2. This is why the HA antibody (3F10) was used to efficiently detect polyubiquitination signals. Polyubiquitinated proteins are less abundant than non-ubiquitinated ones in vivo (cells), but polyubiquitination can be detected by expressing tagged ubiquitin. Hence, it is difficult to detect polyubiquitinated FLAG-MDM2 with a FLAG antibody.
- The results presented in section 3.4. are not conclusive. There is a lot of cullin in the cell, there is no need to transfect it. The authors did not address if the cullin, they transfected, is incorporated in to the SCF complex. It is possible that the interaction, ( competition ) occurs only with free cullin and is not relevant for the SCF complex. One could immunoprecipitate a different subunit of the SCF complex, and detect co immunoprecipitated proteins like, cullin and MDM2, as monomeric cullin will not support any ubiquitin chain formation.
Response: We thank the reviewer for these interesting comments. In section 3.4, we analyzed whether the binding of Cullin1 to MDM2 and also to RPS4X, and further determined their respective binding domains. Indeed, as the reviewer pointed out, this case is expected to show the binding of free Cullin1 to RPS4x and not to Cullin1 the SCF complex. We currently anticipate that RPS4X binding to Cullin1 may reduce the amount of Cullin1 incorporated into the SCF complex and are continuing to analyze the details of this molecular mechanism. If so, it is expected that the SCF complex, including Cullin1, will have a significant impact on the degradation of various proteins targeted by the SCF complex, which will further expand the scope of research in the future.
- The sentence staring in lane 24: “ RPS4X expression in cells enhanced the steady-state level of 24 MDM2 protein. “ is not justified by the results presented. The experiment presented in section 3.5 is very convincing, but it shows a difference in half life, rather than in steady-state level. The values for MDM2 at time 0 with and without RPS4X look very similar. Also the amount of MDM2 in fig 3 B panel 2 and 3 does not show a serious difference with and without RPS4X, the WB on the lysate appears to have even less MDM2 with RPS4X. It seams that the difference in degradation is somehow compensated by higher expression. This may be due to the over expression of the proteins. The half life reported also differs significantly from other reported half life’s of MDM2. Please see: https://www.jbc.org/article/S0021-9258(19)34705-2/pdf; and https://www.ncbi.nlm.nih.gov/pmc/articles/PMC391059/;
These differences may be due to different cell lines and or expression levels. The authors should at least discuss the differences.
Response: We thank the reviewer for this advice. Indeed, our experimental data show that the half-life differs significantly from other half-life reports for MDM2. There are many possible reasons for this discrepancy, as reviewer also pointed out, including differences in cell lines and media conditions. It has also been reported that MDM2 degradation is regulated by various post-translational modifications such as phosphorylation, acetylation, and SUMOylation. As suggested by the reviewer, we described the below to results section (section 3.5) and cited up several references, including reviewer’s recommendations, in the revised manuscript.
“In this result, the half-life of MDM2 seems to differ significantly from previously studies for MDM2 (1, 27). It is well known that the half-life of MDM2 varies widely among cell lines and is also tightly regulated by various post-translational modifications (2). In view of these reports, the half-life of Mdm2 seems to vary greatly depending on the individual experimental conditions.”
1, Peng, Y.; Chen, L.; Li, W.; Agrawal, S.; Chen, J. Stabilization of the MDM2 oncoprotein by mutant p53. J Biol Chem 2001, 276, 6874-6878. DOI:10.1074/jbc.C000781200.
2, Li, J.; Kurokawa, M. Regulation of MDM2 stability after DNA damage. J Cell Physiol 2015, 230, 2318-2327. DOI:10.1002/jcp.24994.
Minor points:
There are no molecular weight markers on any of the WB presented. This makes it difficult to judge the quality.
Response: According to the reviewer’s comment, we have added the molecular weight markers in Figure 2B, 2D, 3B, 4C and 4E.
Reviewer 2 Report
Comments and Suggestions for Authors
Authors in this manuscript showed the regulatory role of RPS4X in promoting MDM2 stability. Although the authors provided convincing data, there are several issues in the manuscript which should be thoroughly addressed by authors.
1. This sentence in not clear in the abstract section- “RPS4X was associated with MDM2 and Cullin1 along with blocking MDM2/Cullin1 formation”. Authors should rephrase it and explain more clearly. Does MDM2/Cullin1 formation means their interaction?
2. In Figure 1A, blots and their corresponding labels are not properly aligned. They should be aligned properly. Also, the WB images are cropped in different sizes. All these should be cropped to have an equal size.
3. In Figure 1A: Authors showed the interaction between overexpressed MDM2 and RPS4X. Did the authors try to do endogenous co-IP to show the interaction between these two proteins? I would recommend the authors to do endogenous co-IP and include that data.
4. In Figure 1A: Did the authors check the effect of RPS4X O/E on the endogenous MDM2 levels? Authors should expect to see an increase in the levels of endogenous MDM2. This data should be included.
5. In Figure 1B: Authors wrote that MDM2 is distributed mainly in the nucleolus. But looking at the image it does not look like that. Authors should replace this image with other image and maybe also provide more enlarged images clearly showing the distribution in nucleolus. Also, authors should include few images in the supplementary data to show that the distribution of MDM2 in nucleolus is not seen in just few cells but is observed in majority of cells.
6. None of the WB images include MW markers in the whole manuscript. I would recommend authors to include molecular weight markers for all the WB images.
7. For figures B and D authors should indicate the MW for all the proteins (full length and deletion mutants) in the figure legends.
8. In the Figure D: Blot for WCL with IB: anti-Myc why there is a strong band in lane 1, which is at the same size as Myc-RPS4X. Although it was not transfected with Myc-RPS4X. Authors should explain?
9. In Figue 3: A and B: Immunoblots are not properly aligned. Authors should align them properly.
10. Page 7, Line 264: The word substantial should be replaced with substantially.
11. In the title 3.4- Authors should write MDMA/Cullin1 complex formation. Include the word complex.
12. Figure 4: C and E, molecular weights for all the deletion domains and full length should be included in the figures, along with MW markers.
13. In Figure 5: The WB should include the blot showing the expression of FLAG-Cullin1. Also loading control is missing. Did the authors normalize the data with loading controls first before making the graph showing the relative protein level compared to control. All these important missing blots and information should be included.
14. Authors mentioned that they did yeast two-hybrid screening assay. I don’t see any data for this assay in the manuscript. This data is missing and should be included.
15. Authors showed all the interaction studies using Co-IP’s using only one cell model i.e. HEK-293T cells. I understand that authors choose this model because this cell line preferred over others because they are easy to transfect and easy to get good quantity of proteins. But I would recommend that authors should also show the interaction between MDM2 and RPS4X in other cancer cell model. This will increase the credibility of data and will show that this interaction occurs across cell types.
16. I would also recommend that in the discussion section authors should include the clinical relevance of this finding. They should discuss how this finding can be used for translational purpose?
Comments on the Quality of English Language
English language is ok. No major corrections are required.
Author Response
We are grateful to reviewer 2 for the critical comments and useful suggestions that have helped us to improve our paper considerably. As indicated in the responses that follow, we have taken all these comments and suggestion into account in the revised version of our paper.
- This sentence in not clear in the abstract section- “RPS4X was associated with MDM2 and Cullin1 along with blocking MDM2/Cullin1 formation”. Authors should rephrase it and explain more clearly. Does MDM2/Cullin1 formation means their interaction?
Response: Thank you for pointing this out. According to the reviewer’s comment, we changed the abstract section of the revised manuscript from “RPS4X was associated with MDM2 and Cullin1 along with blocking MDM2/Cullin1 formation” to “RPS4X associated not only with MDM2 but also with Cullin1, and then blocked the MDM2/Cullin1 interaction” (page1, line 25-26).
- In Figure 1A, blots and their corresponding labels are not properly aligned. They should be aligned properly. Also, the WB images are cropped in different sizes. All these should be cropped to have an equal size.
Response: As suggested by the reviewer, we improved the immunoblot data in Figure 1A in the revised manuscript.
- In Figure 1A: Authors showed the interaction between overexpressed MDM2 and RPS4X. Did the authors try to do endogenous co-IP to show the interaction between these two proteins? I would recommend the authors to do endogenous co-IP and include that data.
Response: We thank the reviewer for this insightful comment. This point was also mentioned by reviewer 1. Combined with the suggestion in the reviewer’s comment 15 to show the interaction between MDM2 and RPS4X in other cancer cell model, we investigated the interaction between endogenous MDM2 and RPS4X in HeLa cells (human cell line derived from cervical cancer) by co-immunoprecipitation assay. As a result, endogenous MDM2/RPS4X interaction was also observed in HeLa cells. This result was added to Figure 1B as a new data and updated the manuscript.
- In Figure 1A: Did the authors check the effect of RPS4X O/E on the endogenous MDM2 levels? Authors should expect to see an increase in the levels of endogenous MDM2. This data should be included.
Response: We thank the reviewer for this comment. According to the reviewer’s suggestion, we re-performed transfection experiment and the western-blot analysis using anti-MDM2 antibody by increasing the amount of expressing vector of RPS4X. Unfortunately, overexpression of RPS4X only slightly promoted endogenous MDM2 stability. This may be due to insufficient efficiency of transfection into the cells, and therefore the effect of RPS4X is not observed in non-transfection cells. To overcome this problem, we believe that it is necessary to construct expression vectors based on viral system, which will enable efficient expression in all cells. However, since the reviewer's comment is valid, we co-transfected RPS4X and MDM2 in cells, and analyzed the stability of MDM2 in the absence of treatment with MG132. As a result, the stability of MDM2 was indeed facilitated by RPS4X. This result was added to Figure 5B as a new data and updated the manuscript.
- In Figure 1B: Authors wrote that MDM2 is distributed mainly in the nucleolus. But looking at the image it does not look like that. Authors should replace this image with other image and maybe also provide more enlarged images clearly showing the distribution in nucleolus. Also, authors should include few images in the supplementary data to show that the distribution of MDM2 in nucleolus is not seen in just few cells but is observed in majority of cells.
Response: We deeply apologize for the confusion and thank the reviewer for this helpful comment. We found a mistake in the previous version of the manuscript. Previously reported that MDM2 is expressed primarily in the nucleoplasm, except for nucleolus (below references 1,2). Similar to these reports, our results showed that MDM2 is distributed throughout the nucleoplasm, except for the nucleolus.
We corrected from “As shown in Figure 1B upper panels, GFP-MDM2 alone was distributed throughout the nucleoplasm, mainly in the nucleolus,” to “Previously reported that MDM2 is expressed primarily in the nucleoplasm, except for nucleolus [1, 2]. Similar to these reports, our results showed that GFP-MDM2 alone was distributed throughout the nucleoplasm, except for the nucleolus.” in the results section (section 3.1), and added these references to the revised manuscript (page 5, line 181-183).
1, Zhang, Y.; Xiong, Y. Mutations in Human ARF Exon 2 Disrupt Its Nucleolar Localization and Impair Its Ability to Block Nuclear Export of MDM2 and p53. Molecular Cell 1999, 3, 579-591. DOI:10.1016/s1097-2765(00)80351-2
2, Weber, J.D.; Taylor, L.J.; Roussel, M.F.; Sherr, C.J.; Bar-Sagi, D. Nucleolar Arf sequesters Mdm2 and activates p53. Nat Cell Biol 1999, 1, 20–26. DOI:10.1038/8991
- None of the WB images include MW markers in the whole manuscript. I would recommend authors to include molecular weight markers for all the WB images.
Response: We appreciate the reviewer’s recommendation on this point and we agree. Unfortunately, in some blots other than ubiquitin or deletion mutants, there were present between the MW markers. Therefore, we could label only the panel about ubiquitin and deletion mutants.
- For figures B and D authors should indicate the MW for all the proteins (full length and deletion mutants) in the figure legends.
Response: According to the reviewer’s comment, we have added the MW of full-length and deletion mutants of MDM2 and RPS4X in Figure 2B, and 2D. Generally, the molecular weight (kDa) of a protein is derived from the degree of migration of the molecular weight marker in the gel. Therefore, for individual proteins, it is not possible to give their exact molecular weights as numbers, and we do not believe that this is common.
- In the Figure D: Blot for WCL with IB: anti-Myc why there is a strong band in lane 1, which is at the same size as Myc-RPS4X. Although it was not transfected with Myc-RPS4X. Authors should explain?
Response: We deeply apologize to the reviewer for this dirty panel. It seems that due to a mistake in the procedure, the sample of lane 2 leaked to lane 1. Therefore, we re-performed the same experiment and showed as a new data of the 2nd and 4th panels in Figure 2D.
- In Figue 3: A and B: Immunoblots are not properly aligned. Authors should align them properly.
Response: In accordance with the reviewer’s comment, we modified Figure 3A and 3B.
- Page 7, Line 264: The word substantial should be replaced with substantially.
Response: We thank the reviewer for pointing out this mistake. According to the reviewer’s comment, we corrected from “Since the interaction between MDM2 and Cullin1 was substantial attenuated by RPS4X, we expected that RPS4X would interact with both MDM2 and Cullin1.” to “Since the interaction between MDM2 and Cullin1 was substantially attenuated by RPS4X, we expected that RPS4X would interact with both MDM2 and Cullin1.” in the results section to the revised manuscript (page 9, line 301-302).
- In the title 3.4- Authors should write MDMA/Cullin1 complex formation. Include the word complex.
Response: According to the reviewer’s comment, we changed the title 3.4. as follows: “RPS4X associates with the N-terminus of Cullin1 through its C-terminus and potently inhibits MDM2/Cullin1 complex formation”.
- Figure 4: C and E, molecular weights for all the deletion domains and full length should be included in the figures, along with MW markers.
Response: According to the reviewer’s comment, we added the MW markers in Figure 4C and 4E.
- In Figure 5: The WB should include the blot showing the expression of FLAG-Cullin1. Also loading control is missing. Did the authors normalize the data with loading controls first before making the graph showing the relative protein level compared to control. All these important missing blots and information should be included.
Response: Thank you for pointing this out. According to reviewer’s suggestion, we added the blots of FLAG-Cullin1 and β-actin as loading controls. We also normalized the MDM2 protein levels with β-actin and plotted the graph again.
- Authors mentioned that they did yeast two-hybrid screening assay. I don’t see any data for this assay in the manuscript. This data is missing and should be included.
Response: We thank the reviewer for this comment and described the below to results section (page 5, line169-172) in the revised manuscript.
“To screen for proteins that interact with the central region of MDM2, we used the yeast two-hybris system. A screen of 4 x 105 primary transformants of universal human cDNA library yielded 165 potential positive clones. Of these candidates, sequence analysis revealed the several positive clones corresponding to RPS4X.”
- Authors showed all the interaction studies using Co-IP’s using only one cell model i.e. HEK-293T cells. I understand that authors choose this model because this cell line preferred over others because they are easy to transfect and easy to get good quantity of proteins. But I would recommend that authors should also show the interaction between MDM2 and RPS4X in other cancer cell model. This will increase the credibility of data and will show that this interaction occurs across cell types.
Response: This comment was discussed in our response 3. Please see response 3.
- I would also recommend that in the discussion section authors should include the clinical relevance of this finding. They should discuss how this finding can be used for translational purpose?
Response: We thank the reviewer for this helpful comment and totally agree. As suggested by the reviewer, to emphasize the clinical relevance of our current findings, we changed to below and cited up two references (below references 3,4), in the revised manuscript (page 11, line 358-367).
“A previous study suggested that overexpression of RPS4X is correlated with poor prognosis in not only colon rectal cancer but also intrahepatic cholangiocarcinoma [3]. Additionally, overexpression of MDM2 protein was observed in 38% of intrahepatic cholangiocarcinoma tumors and correlated with the presence of metastases (P<0.01) and advanced tumor stage (P<0,05) [4]. In the view of these reports and our findings in the present study, enhancement of MDM2 protein stability by RPS4X potentially could lead to tumor progression and poor prognosis. Further studies are required to elucidate the functional consequences of the interaction between RPS4X and MDM2 in cancer cells and may contribute to reinforcing the evidence that it could be a biomarker for these cancers, which may help in more effective treatment for patients.” (page 11, line 358-367). We hope that our revision is reasonable for the reviewer.
3, Kuang, J.; Li, Q.Y.; Fan, F.; Shen, N.J.; Zhan, Y.J.; Tang, Z.H.; Yu, W.L. Overexpression of the X-linked ribosomal protein S4 predicts poor prognosis in patients with intrahepatic cholangiocarcinoma. Oncol Lett. 2017, 14, 41-46. DOI:10.3892/ol.2017.6137
4, Horie, S.; Endo, K.; Kawasaki, H.; Terada, T. Overexpression of MDM2 protein in intrahepatic cholangiocarcinoma: relationship with p53 overexpression, Ki-67 labeling, and clinicopathological features. Virchows Arch. 2000, 437, 25-30. DOI: 10.1080/15384101.2015.1120926
Round 2
Reviewer 1 Report
Comments and Suggestions for Authors There is still one critical point.In my first review i wrote:
The authors used HA-tagged Ubiquitin for the experiment in fig. 3B, to show less ubiquitinylation of MDM2. This is not conclusive, as the is no control showing equal expression or transfection of HA-Ubiquitin.
The authors explained in their point to point reply, why they transfected HA-Ubiquitin, but the transfection control is still missing, and no explanation offered.
All other point are solved.
Author Response
We are grateful to reviewer #1 for the critical comment and useful suggestion that have helped us to improve our paper considerably. As indicated in the response that follow, we have taken this comment and suggestion into account in the revised version of our paper.
- There is still one critical point. In my first review i wrote:
The authors used HA-tagged Ubiquitin for the experiment in fig. 3B, to show less ubiquitinylation of MDM2. This is not conclusive, as the is no control showing equal expression or transfection of HA-Ubiquitin. The authors explained in their point to point reply, why they transfected HA-Ubiquitin, but the transfection control is still missing, and no explanation offered.
All other point are solved.
Response: We deeply apologize for the confusion and thank the reviewer for pointing this out. According to the reviewer’s comment, we added the blot of HA-Ubiquitin of whole cell lysate in Figure 3B. Interestingly, when RPS4X is co-expressed in the cells, the level of ubiquitinated proteins are rather high. This is probably due to enhanced ubiquitination of cell-substrate proteins by MDM2 via stabilization of MDM2.
Reviewer 2 Report
Comments and Suggestions for Authors
Authors addressed all my comments.
Author Response
We truly thank you for your careful review of our manuscript and constructive comments.
Authors addressed all my comments.
Response: We kindly appreciate the effort put on the careful revision of this manuscript.